# Stem Cell Therapy in Diabetic Polyneuropathy: Recent Advancements and Future Directions

**DOI:** 10.3390/brainsci13020255

**Published:** 2023-02-02

**Authors:** Shamima Akter, Mayank Choubey, Mohammad Mohabbulla Mohib, Shahida Arbee, Md Abu Taher Sagor, Mohammad Sarif Mohiuddin

**Affiliations:** 1Department of Internal Medicine, St. Francis—Emory Healthcare, 2122 Manchester Expressway, Columbus, GA 31904, USA; 2Department of Foundations of Medicine, NYU Long Island School of Medicine, 101 Mineola Blvd, Mineola, NY 11501, USA; 3Julius Bernstein Institute of Physiology, Medical School, Martin Luther University of Halle-Wittenberg, Magdeburger Straße 6, 06112 Halle, Germany; 4Institute for Molecular Medicine, Aichi Medical University, 1-Yazako, Karimata, Aichi, Nagakute 480-1103, Japan; 5Department of Pharmacology, School of Medical Sciences at UNSW Sydney, Sydney, NSW 2052, Australia

**Keywords:** diabetic polyneuropathy, cytotherapy, stem cell, stem cell therapy, degenerative diseases

## Abstract

Diabetic polyneuropathy (DPN) is the most frequent, although neglected, complication of long-term diabetes. Nearly 30% of hospitalized and 20% of community-dwelling patients with diabetes suffer from DPN; the incidence rate is approximately 2% annually. To date, there has been no curable therapy for DPN. Under these circumstances, cell therapy may be a vital candidate for the treatment of DPN. The epidemiology, classification, and treatment options for DPN are disclosed in the current review. Cell-based therapies using bone marrow-derived cells, embryonic stem cells, pluripotent stem cells, endothelial progenitor cells, mesenchymal stem cells, or dental pulp stem cells are our primary concern, which may be a useful treatment option to ease or to stop the progression of DPN. The importance of cryotherapies for treating DPN has been observed in several studies. These findings may help for the future researchers to establish more focused, accurate, effective, alternative, and safe therapy to reduce DPN. Cell-based therapy might be a permanent solution in the treatment and management of diabetes-induced neuropathy.

## 1. Introduction

In the modern world, the number of patients being affected by chronic metabolic disorders, including obesity, metabolic syndromes, dyslipidemia, and diabetes, is increasing. It is assumed that high-calorie intake, a sedentary lifestyle, and the consumption of fructose-containing beverages and foods lead to these metabolic disorders [1]. Diabetes occurs around the world and is more common in developed countries [2,3,4]. In the western world, diabetic polyneuropathy (DPN) (Box 1) is recognized as the most common complication related to diabetes. It has been estimated that 10–100% of patients with diabetes are affected by clinical or subclinical neuropathies [5]. Several studies have said that approximately 50% of the patients with diabetes would eventually develop polyneuropathy [6,7,8]. Additionally, DPN causes diabetic feet, including foot infections, ulcers, and limb amputations. It has been reported that at least 15% of diabetes will develop a foot ulcer [9].

Box 1Definition and Classification of DPN.Definition: According to a recent statement from the American Diabetes Association, DPN is defined as, “the presence of symptoms and/or signs of peripheral nerve dysfunction in people with diabetes after the exclusion of other cause” [10].Classification of DPN [11]1. Typical diabetic polyneuropathy or sensorimotor polyneuropathy (DSPN)-Primary small fiber-Subclinical DSPN2. Atypical diabetic neuropathy3. Painful DPN

Worldwide, tight blood glucose control is the only accepted therapeutic strategy to prevent the development of DPN, especially in type 1 diabetes [12,13,14]. In the pharmacological approach, an aldose reductase inhibitor is being used clinically in Japan and India. Alpha lipoic acid and benfotiamine are licensed for clinical use for treating DPN. However, there is no curable treatment in the progressive stage of DPN [15,16]. Two ideal strategies have been proposed for the curative treatment of DPN; one is gene therapy, and another is cell therapy [17,18].

For over a decade, stem cell therapy has been used as a therapeutic for different types of diseases due to its novel approach. The robust potential of stem cells to differentiate into specific types of cells and regenerate tissues and body organs has been proven in several studies [19]. Stem cell therapy has been believed to be a promising regenerative therapy for different neurological diseases including DPN because of its potency of regeneration and paracrine secretion of several factors such as angiogenic and neurotrophic factors (Refer to Table 1) [20]. In this article, we will discuss the possibility and future of cell therapy for the cure of progressive DPN.

## 2. Clinical Manifestations of DPN

DPN is primarily distal symmetrical sensory polyneuropathy affecting the distal lower extremities. In most patients, symptoms of polyneuropathy can be described as “positive and negative symptoms.” Positive symptoms are superficial burning, paresthesia, deep aching pains, dysesthesia, contact-induced discomfort, and paroxysmal jabbing pains. These worsen at nighttime [51]. Negative symptoms are the loss of sensations to touch, vibration, pinprick, hot, and cold [51,52,53].

## 3. Etiology of DPN

To date, the exact mechanism of DPN remains unclear. However, hyperglycemia is generally known as the primary cause of DPN in type 1 and 2 diabetes. The recent literature suggests a number of potential pathways that may contribute to the development of DPN (Figure 1). Excess intracellular accumulation of glucose leads to reactive oxygen species (ROS) [54], damage to microvasculature [55], diminished neurotrophic factors, impaired nerve blood flow, reduced neuronal integrity [56,57], reduced nerve conduction velocity, and nerve energy failure [58]. Additionally, dyslipidemia occurs mainly in type 2 diabetes and is linked to DPN [59]. Growth factors such as vascular endothelial growth factor (VEGF), insulin-like growth factor (IGF), nerve growth factor (NGF), brain-derived neurotrophic factor (BDNF), and fibroblast growth factor-2 (bFGF) have both neurotrophic and angiogenic effects [60]. In diabetic mellitus (DM), all these components are diminished, causing functional atrophy and even nerve cell death, which concludes in the etiology of DPN [52,61]. Inflammatory mediators such as NF-κB, and TNF-α, TGF-β are produced as a result of the various glucose-induced pathways that induce oxidative stress and myelin damage [8,62,63]. Insulin receptors have been expressed in the dorsal root ganglion and found the neurotrophic effects of insulin in the peripheral nerves [64]. Insulin exerts its neurotrophic effects, the promotion of neuronal growth and survival, and insulin resistance leads to reduced neurotrophic signaling that contributes to the pathogenesis of peripheral nerves [65].

Collectively, the biochemical damage induced by advanced glycation end products results in impaired nerve blood flow and diminished neurotrophic support, [15] and may play a role in disrupting neuronal integrity and repairing the mechexosamine pathway [66].

## 4. Current Management of the DPN

### 4.1. Prevention

According to the above statement, hyperglycemia and insulin deficiency played an essential role in the pathogenesis of DPN. Glycemic control and lifestyle modifications are the main focuses for preventing DPN [14,67].

### 4.2. Glucose Control

With type 1 diabetes, 78% of patients reduce the incidence of diabetes by enhancing tight glucose control [68]. However, 5–9% reduces the relative risk of DPN by controlling glycemic states [69,70]. Intensive insulin therapy can prevent or delay the development of DPN [71]. Lynn et al. observed that still today, tight glycemic control is the only content strategy that showed delay or prevention of DPN in type 1 patients with diabetes and slows the progressive development of neuropathy in some patients with type 2 diabetes [72]. Maintaining a stable blood glucose level reduces the symptoms of neuropathy and further nerve damage by 50% [73]. Anti-diabetic drugs such as Metformin and Thiazolidinediones (TZD) group can also play a role not only for controlling the glucose levels but also have beneficial vascular effects. Metformin prevents oxidative stress-induced cell death [74] and has neuroprotective effects via the inhabitation of oxidative stress-related neuronal cell death [75]. However, TZDs reduce inflammation and improve endothelial dysfunction [76] and oxidative stress [77]. Wiggin et al. observed that treatment with rosiglitazone, a TZD, reduces DN in streptozotocin (STZ)-treated DBA/2J mice. Rosiglitazone reduces oxidative stress and prevents the development of thermal hypoalgesia [78].

### 4.3. Pharmacological Approach

DPN can reduce the quality of life and enhances depression and social dysfunction [79]. A pharmacological approach is needed for DPN beyond lifestyle management or glycemic control [80,81]. There are four categories of drugs are available for DPN: 1. anticonvulsants (particularly alpha-2-delta ligand); 2. antidepressants (mainly TCAs); 3. serotonin-norepinephrine reuptake inhibitors (SNRIs); 4. opiate-receptor agonists and topical agents. Alpha-2-delta ligand: Pregabalin is one of the most used drugs for DPN. It received regulatory approval from the FDA, Health Canada, and the European medicine agency for treating DPN. It shows a 30–50% improvement in pain due to DPN [82,83,84,85,86,87]. However, not all trials with pregabalin show positive effects [82,83,88,89]. TCAs: Amitriptyline shows greater efficacy in painful DPN than other TCAs [90]. It needs a trial of 6–8 weeks to assess its effect [91]. However, FDA unapproved TCAs for treating DPN [92]. SNRIs: Duloxetine shows efficacy for treating painful DPN in multicenter randomized trials [82,83,87,88,89]. A small increase in HbA1c was observed in the patients treated with duloxetine. Tramadol, a topical capsaicin, may be useful and should be considered for treating painful DPN. Mohiuddin et al. reported that glucagon and glucagon-like peptides have some beneficial effects on the DPN *in vitro* model [93,94,95]. However, this is not yet accepted [82,83,96]. Despite the fact that all of these DPN medications may lessen the symptoms, none of the studies shows effectiveness to prevent the progression of DPN. 

### 4.4. Angiogenic and Neurotrophic Factor Therapy

Neurotrophic factors such as NGF [52], IGF1 [97], and IGF2 [98] and ciliary neurotrophic factor (CNTF) [99] or glial cell line-derived neurotrophic factor (GDNF) have been shown to ameliorate DPN in animal models [100]. VEGF shows benefits to improve DPN at a certain level [101]. Intramuscular administration of FGF-2 increased blood flow in the sciatic nerve and improved nerve conduction velocity [102]. Moreover, prevention and minimization of metabolic disturbances, vascular damages, neuronal cell injuries, nerve perfusion, and ischemia may be approachable for DPN [52].

## 5. Stem Cell Therapy in DPN

In several studies, stem cell therapy appears very promising for the treatment of DPN. Stem cells can differentiate into the cells that are necessary to repair the damaged peripheral nerves and blood vessels. Adult stem cells and growth factors are injected into the damaged areas to reduce pain and improve blood flow to nerves. In our review, we have discussed various types of stem cells with their mechanism that is involved in the treatment of DPN [19].

### 5.1. Bone Marrow Mononuclear Cell Therapy

Bone marrow-derived cell therapies are the most accepted therapies because of their unique nature. An advantage of using circulating or BM-derived cells is that they can be harvested from a patient’s bone marrow and re-introduced back to the patient [103,104]. Thus, there is no chance of graft rejection. Bone marrow (BM) is a source of mononuclear cells (MNC). The BM-MNC is the term used to entitle all the cells in the bone marrow with unilobulated or rounded nuclei and inadequacy of granules in the cytoplasm [105]. Bone marrow-derived mononuclear cells (BM-MNCs) are a heterogeneous group of cells, which include mainly endothelial progenitor cells (EPCs), mesenchymal stromal cells (MSCs), and hematopoietic stem cells (HSCs) [106].

An advantage of using BM-MNCs as the source of cell therapy is that they are rather easy to acquire. They can be isolated from bone marrow by centrifugation and do not require the ex vivo culture system. Some studies have shown the beneficial effects of using BM-MNCs in the case of DPN. They improved neovascularization by increasing the levels of angiogenic factors such as VEGF, FGF-2, and angiopoietin-1 (Figure 2) [107,108]. In patients with ischemia, BM-MNCs transplantation has also been reported to be beneficial [108]. Because BM-MNCs transplantation has been shown to be an option in treating ischemic diseases, there was an interest in using a similar strategy in treating DPN [109]. A recent study showed that peripheral blood mononuclear cell implantation in rats with DPN partially recovered blood flow and improved the motor nerve conduction velocity (MNCV) of the sciatic nerve [110]. Kim, Park, Choi et al. reported that intramuscular transplantation of BM-MNCs enhances the expression of a number of angiogenic and neurotrophic factors, including VEGF, FGF-2, IGF-1, and NOS-3, in the vasa nervorum of DPN model rats. As a result, it improved nerve conduction velocity and promotes nerve vascularity. [111]. Shibata et al. reported that transplantation of BM-MNCs in STZ-induced 8-week age diabetic rat improves the transplantation of NCV, improves sciatic nerve blood flow, and increased the density of small vessels in the muscle. However, BM-MNCs taken from age-matched patients cannot show any beneficial effects in DPN [112]. Kondo et al. reported that transplantation of BM-MNCs derived from young rats ameliorated DPN, but BM-MNCs from mature or diabetic rats cannot show any efficacy [113]. Despite the beneficial effects of MSC transplantation in experimental DN shown previously, there appears to be a significant limitation in using MSCs for DPN therapy. A study showed that BM-derived MSCs might undergo chromosomal abnormalities and form malignant tumors after injection into mice with DPN. This study describes the careful monitoring of chromosomal status for transplantation of MSCs from *in vitro* expansion [114].

### 5.2. Pluripotent Stem Cell Therapy

Pluripotent stem cells are derived artificially from the mature somatic cell by insertion of (oct3/4, sox2, klf4, c-myc) using retrovirus as a vector. However, there is a chance of tumor formation because the c-myc is a potent oncogene [115,116]. Pluripotent stem cells are the new hope for regenerative medicine as they can produce every cell type in the body. Induced pluripotent stem cells (iPS) are used for organic synthesis, such as in the liver from human ‘liver buds’ iPSCs-LBs [117], and tissue repair, such as when iPSs are injected into the vitreous of damaged retina and the iPSCs engrafted into the retina [118]. Impaired vascularity and nerve degeneration are the most crucial pathophysiology of DPN. The neural crest-like cells (NCL) derived from iPS may have a therapeutic effect on DPN [119]. Angiogenesis occurs in STZ-induced diabetic mice by transplantation of NCL. It is due to the action of the angiogenic factor, vascular endothelial growth factor (VEGF) and basic fibroblast growth factor (bFGF), NGF, and Neurotrophin-3, which are secreted by NCL. Okawa et al. reported that transplantation of NCL derived from the aged mice into the 16 weeks STZ-induced mice. After 4 weeks, these transplanted cells produced growth factors such as NGF and Neurotrophin-3 and differentiated into vascular smooth muscle a cells which improve the impaired nerve and vascular functions [118]. However, it is questionable how long the locally grafted iPS cells will respond and in which amount there is no chance of tumor formation. Obstacles such as the chance of tumor formation, epigenetic memory, and new features obtained during remodeling are seen in iPSCs [120]. However, if these obstacles can be removed then, pluripotent stem cell therapy can be a good option for treating advanced stage DPN.

### 5.3. Endothelial Progenitor Cells (EPCs) Therapy

EPCs exhibit various therapeutic uses. These cells can also be isolated from umbilical cord blood and peripheral blood. EPCs can differentiate within endothelial cell intima of existing blood vessels. EPCs can be identified using several markers such as VE-Cadherin, CD31, CD34, CD45, CD45.1, CD45.2, CD117, CD133, CXCR4, ER71, CD146, Tie-2, VEGF-R2, and VEGF-R3. However, the majority of studies suggest CD 34+/KDR+/CD133+ markers for identifying EPCs [121,122]. Some studies have reported direct augmentation of neural neovascularization in the sciatic nerves of mice with DPN after local intramuscular injection of BM-derived EPCs [123]. The injected EPCs preferentially went to peripheral nerves; on the other hand, they went much less to the muscles. This shows that muscular neovascularization is not the mechanism at work. Additionally, the study showed that EPCs have durable engraftment into diabetic nerves [124]. Naruse et al. reported that intramuscular injection of EPCs in the hind limbs of STZ-induced diabetic nude rats increased the differentiation of endothelial cells in the hind limb. This results in the improvement of sciatic nerve conduction velocity (NCV) and blood flow [117]. However, this study is unable to demonstrate the exact mechanism through which EPCs enhance vascular health. Some of the recent studies did not agree about that EPCs differentiation as a major mechanism for neovascularization [125,126]. However, the main therapeutic effects are not through endothelial differentiation but are through angiogenesis; the overall evidence clearly suggests that BM-derived EPCs take part in blood vessel formation through vasculogenesis. Although the differentiation of EPCs plays a vital role in the recovery of damaged tissue, function is still controversial. Some studies have shown that the differentiation of EPCs into endothelial lineage cells, and they are incorporated into blood vessel formation [[122,123]. Naruse et al. reported that transplantation of cord blood-derived EPCs into the hind limb skeletal muscle increased the differentiation of EPCs into the endothelial cell in soleus muscle that leads to increase sciatic motor nerve conduction velocity and sciatic nerve blood flow in a rat DPN model [117]. However, this study does convey any clear idea about the mechanism by which the transplanted EPCs increase neovascularization or nerve conduction velocity. More recent studies have argued against the fact that EPCs do not differentiate into ECs [127,128].

### 5.4. Mesenchymal Stromal Cells Therapy

Mesenchymal stromal cells (MSCs) are the multipotent cells generally found in almost all post-natal organs and tissues [129] such as the umbilical cord [130], placenta [130], and dental pulp [131], and all of these have differentiation properties, which is multipotent. Meanwhile, MSCs are stupendous candidates for treating DPN. MSCs are identified by several markers such as CD54/CD102, CD166, CD73, CD90, CD44, and CD105 [20]. MSCs can differentiate into mesodermal cells in origin, such as bone, cartilage, and adipose tissue [132]. However, there is a controversy of whether MSCs are true stem cells or not, as they cannot fulfill the criteria of a stem cell. Their proliferation is self-limited, and self-renewal capacity of human MSCs still unproven [133,134,135]. H. J. Park et al. reported that transplantation of hMSCs into a Parkinson’s disease rat model exerted a neuroprotective effect [136]. Until January 2023, according to the statement of https://clinicaltrials.gov (accessed on 16 December 2022), almost 1158 clinical trials have been conducted on MSCs. However, most of these are the phase I or II that evaluate the effectiveness of MSCs in hepatic, nervous, renal, bone/cartilage, or autoimmune disorders. MSCs secrete anti-inflammatory, antiapoptotic molecules and trophic factors such as FGF, VEGF-A, and NGF, which support the growth of axons, remyelination, angiogenesis, and protection from apoptotic cell death through a paracrine effect [112,137]. Siniscalco et al. reported that after transplantation of MSCs in the cerebral ventricle, mechanical allodynia and thermal hyperalgesia are reduced in neuropathic mice [[138]. MSCs also improve the glycemic status by the regeneration of pancreatic beta cells in STZ-induced diabetic mice [139]. However, currently, MSCs have been used against acute graft-versus-host disease in Japan, New Zealand, and Canada.

### 5.5. Dental Pulp Stem Cell Therapy

Dental pulp stem cells (DPSCs) are mesenchymal stem cells located in the dental pulp cavity. DPSCs show high expression of CD29, CD90, and CD49d markers; these are common MSCs markers [140]. DPSCs have the natural capacity to proliferate, and they can differentiate into odontoblasts, adipocytes, osteoblasts, and neuronal cells [131,141]. However, DPSCs are an adorable candidate for cell therapy as they are easy to obtain from tooth extraction from early-age donors without any invasive procedure. DPSCs are highly expressed to neurotrophic and angiogenic factors such as NGF, NT-3, VEGF, and bFGF [41]. DPSC transplantation increases these factors and improves DPN [52,142,143]. Omi et al. reported that transplantation of DPSCs into the hind limb of the skeletal muscles of a rat model of DPN improved the sciatic motor/sensory nerve conduction velocity and sciatic nerve blood flow. They also suggested that transplantation of DPSCs eased the hypoalgesia in the DPN rats. Long-term diabetic nerves show reductions in the fiber area, occupancy rate, myelin area, and myelin thickness. The transplantation of DPSCs increases myelin thickness and myelin area, which indicates effective results for Schwann cells [140].

### 5.6. Embryonic Stem Cell Therapy

Embryonic stem cell therapy (ESC) is one of the possible candidates for treating DPN because it is infinitely renewable and amenable to molecular manipulation capacity [144]. Embryonic stem cells are derived from the pre-implantation of a blastocyst. It possesses an inner cell mass (ICM), which subsequently forms the ES cells, and the outer layer consists of cells collectively called trophoblasts, which give rise to the placenta. One of the most critical aspects of ES cell lines is that in response to appropriate stimuli, they can differentiate into multiple mature somatic cell types representing all three germ layers, both in vivo and *in vitro*. A group of studies shows that hESC can be differentiated into neural crest cells (NCCs) and cells with the morphological and molecular characteristics of myelinating Schwann cells [145,146,147]. ESCs show high expression of Nanog, GTCM 1, connexin 43(GJA1), Oct 4, and TDGF1 (crypto) markers [148]. Nakazawa et al. stated that differentiated NCCs increase several trophic factors, mainly BDNF, which is essential for neuronal survival and the elongation of the axon [149]. Jones et al. reported that transplantation of NCCs derived from ESC showed biologically active trophic factors and could stimulate neurite outgrowth in a rat sciatic nerve injury model [150]. However, no available research shows the use of ESC for treating DPN. Still, now there is no clinical trial for embryonic stem cell therapy in humans due to strong objections by certain religious communities [151,152] and due to the risk of the formation of tumors in vivo [153].

## 6. Challenges in Cell Therapy

Still today cell therapies have been applied only to animal models, and most of the reports have given a positive attitude for treating DN. However, there are particular challenges that have been observed during these studies. Before starting a human trial, these challenges must be faced: 1. risk of tumor formation, 2. graft rejections, 3. the optimal dose for cell survival and to reach the necessity, 4. route of transplantation, 5. the outcome of the transplantation, and 6. details of the mechanism of actions. Sarcoma has been reported in murine MSCs *in vitro* [154] as well as tumor development after allogeneic transplantation of MSCs [155] and BMSC transplantation in STZ-induced diabetic mice. [114].

## 7. The Route of Transplantation

For the efficacy and viability of the transplanted cell, it is essential to consider the mood of transplantation, whether it will be topical, intraocular, or systemic. Without a proper dose and route of transplantation, the efficacy will be reduced or not be observed. For treating diabetic foot ulcers, the administration of BM-MSCs as a nonvascular injection is mostly used currently. Systemic delivery may be one of the most provocative routes of administration. However, this will require a large number of cells to reach the target tissue, and the efficacy of the transplanted cells may be reduced [156].

## 8. Conclusions and Future Directions

As diabetes is a chronic metabolic disorder, it affects human health from head to toe [157,158,159,160,161]. Foot ulceration and limb amputations are the consequences of DPN if there is no effective clinical treatment. Conventional drug therapies are showing limited activities against DPN. Complimentary treatment often shows a potential role. However, its use is still limited. Pipelines do not suggest an active molecule against DPN. Cell therapy may not be a standard treatment option for all stages of DPN because of the different structural or functional changes marked in various stages of DPN. However, the costs and availabilities of these approaches are out of reach for ordinary people. Immunity has also been a problem with direct cell-derived therapies. Delivery approaches must be taken care with proper instrumentation. Though cell-based treatment approaches have been proven well in both animal and human subjects, appropriate clinical evaluations are necessary before making them for general use.

## Figures and Tables

**Figure 1 brainsci-13-00255-f001:**
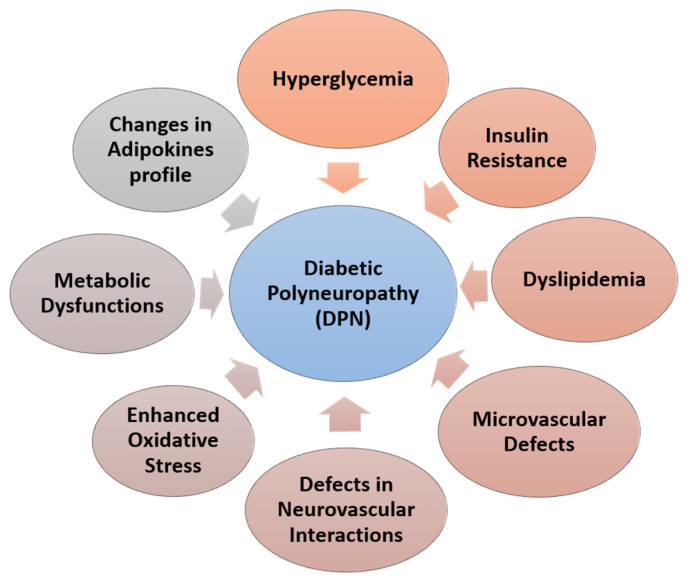
Factors responsible for diabetic polyneuropathy (DPN).

**Figure 2 brainsci-13-00255-f002:**
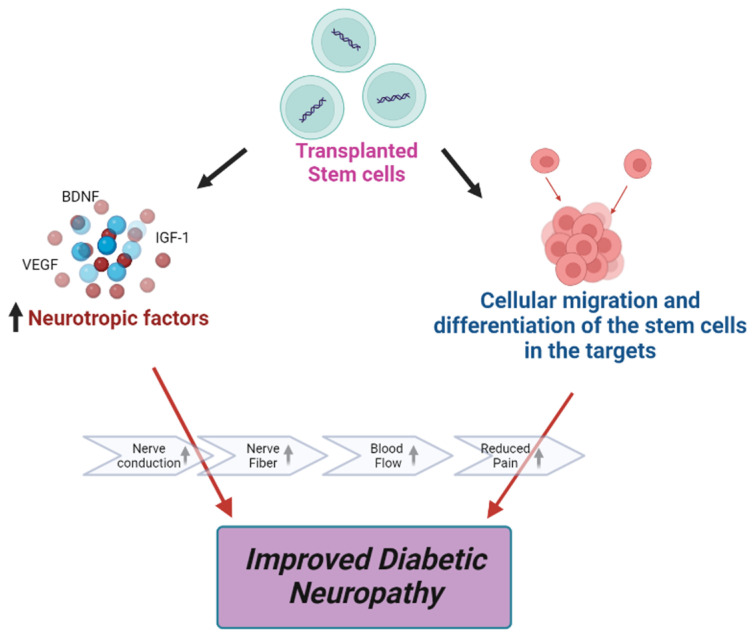
Effects of stem cell therapy on DPN.

**Table 1 brainsci-13-00255-t001:** A review of stem cell therapy and its effects on neurological diseases.

Disease	The Type of Stem Cell	Effect/Mechanism of Action	Ref.
*Alzheimer’s disease*	NSC	Reduced neuroinflammation. Increased neurogenesis, cognitive function, and synaptogenesis, by secretion of neuroprotective agents. Enhance secretion neurogenerative growth factors such as vascular endothelial growth factor (VEGF), nerve growth factor (NGF), Brain derived neurotrophic factor (BDNF), and insulin growth factor-1 (IGF-1).	[21,22,23,24,25,26,27,28,29,30]
	ESC	Continuous production of cholinergic neurons Cognitive function restoration	[31]
	MCSC	Improvement in neuronal differentiation, neurogenesis, synaptogenesis Improvement in locomotor and cognitive functions	[32,33,34]
*Amyotrophic Lateral Sclerosis*	PSC	Differentiate into motor neurons	[35]
	ESC	Differentiate into motor neurons	[35]
	MSC	Neuroprotective effects Increases muscle strength	[35]
	NSC	Reduction of progression of diseases Increases muscle strength Neuroprotective effects	[35]
	PSC	Differentiate into motor neurons	[36]
*Parkinson disease*	HFSC	Re-innervation of the affected areas by dopaminergic actions.	[37]
	NSC	Reproduction of neurons by dopaminergic actions	[38,39]
*Huntington’s disease*	NSC	Differentiation of progenitor cells into neural cells	[38]
		Reduction of degeneration	[40]
	HFSC	Improvement in the behavior	[41]
Stroke	MSC	Improvement of movement	[42]
	NSC	Neuroprotection	[42,43]
	ESC	Recovery from the disease and improved movement Neuroprotection	[44,45]
Spinal cord injury	ESC	Recovery from injury	[46]
		Multipotent neural precursor formation	[47]
	NSC	Re-myelination	[48]
		Promote neuroprotection	[49]
		Recovery from injury	[50]

## Data Availability

Not applicable.

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
