# Peer review of "Stem Cell Therapy in Diabetic Polyneuropathy: Recent Advancements and Future Directions"

_brainsci, 2023, doi:10.3390/brainsci13020255_

Round 1

Reviewer 1 Report

The authors present an interesting review examining the therapeutic potential of stem cell therapy in the context of diabetes, specifically polyneuropathy. Stem cells have represented a valuable opportunity as an effective form of treatment in the context of a number of disease states, however, controversies surrounding their source and use has marred development of practises using such. However, as evidence emerges and the techniques evolve to overcome these challenges, the potential of stem cells in treating aspects of disease like diabetes are gathering pace, and this review eloquently and comprehensively captures the state-of-the-art in terms of how these therapy is evolving in the context of polyneuropathy.

In reviewing the article I made a number of observations. The following should be considered when preparing a suitable revision.

1.       Throughout the manuscript there are instances of non-scientific language and perhaps even typos in some instances. For example, in the abstract cell therapy is described as a ‘spanking candidate’. These instances, of which there are a few, require attention to bring the language towards the scientific publication standard.

2.       The authors utilise a ‘box’ to deliver a definition. I would question whether this is needed or not. Could this information not be integrated into the main body of text?

3.       The authors never make reference to any tables or figures within the text. The authors need to refer the reader to these figures where appropriate.

4.       There appears to possibly be an issue with Section 6 ‘Challenges in cell therapy’. I believe an additional Section 7 may have been created by accident, and this may form part of Section 6 – but perhaps not. If this is not the case, then Section 6 needs to be developed more, and Section 7 needs to be revised in terms of the length of title, and the content within it.

Author Response

Response to reviewer 1 comments:

The authors present an interesting review examining the therapeutic potential of stem cell therapy in the context of diabetes, specifically polyneuropathy. Stem cells have represented a valuable opportunity as an effective form of treatment in the context of a number of disease states, however, controversies surrounding their source and use has marred development of practises using such. However, as evidence emerges and the techniques evolve to overcome these challenges, the potential of stem cells in treating aspects of disease like diabetes are gathering pace, and this review eloquently and comprehensively captures the state-of-the-art in terms of how these therapy is evolving in the context of polyneuropathy.

Thank you very much for raising the important concern that can make this manuscript more suitable.

In reviewing the article I made a number of observations. The following should be considered when preparing a suitable revision.

  1. Throughout the manuscript there are instances of non-scientific language and perhaps even typos in some instances. For example, in the abstract cell therapy is described as a ‘spanking candidate’. These instances, of which there are a few, require attention to bring the language towards the scientific publication standard.

Answer: Thank you for finding this, we have changed that non-scientific word with better scientific words.

  1. The authors utilise a ‘box’ to deliver a definition. I would question whether this is needed or not. Could this information not be integrated into the main body of text?

Answer: Diabetic polyneuropathy is different from other types of diabetic neuropathy. Therefore, it is important to highlight the exact definition and classification for a better understanding of the article. That is why we think it is better to highlight the definition and classification in the box. However, if the reviewer think we can remove it.

  1. The authors never make reference to any tables or figures within the text. The authors need to refer the reader to these figures where appropriate.

Answer: Thank you for noticing this error. We have cited the figures and tables in the text as suggested by reviewer.

  1. There appears to possibly be an issue with Section 6 ‘Challenges in cell therapy’. I believe an additional Section 7 may have been created by accident, and this may form part of Section 6 – but perhaps not. If this is not the case, then Section 6 needs to be developed more, and Section 7 needs to be revised in terms of the length of title, and the content within it.

Answer: Thank you for finding this typo error. We have corrected this line in the text.

Reviewer 2 Report

The manuscript reviewed existing literatures regarding stem cell therapy for diabetic polyneuropathy (DPN). Overall, the manuscript topic is interesting in the field, but it needs improvement.

Specific comments:

1. The manuscript does not seem to be well structured. For example, table 1 on page 2 and 3 lists existing stem cell therapies on neurological diseases other than DPN, but this table is not mentioned in the text of the manuscript at all.

2. I think the manuscript will be improved if the authors could elaborate some more on the mechanisms and beneficial effects of stem cell therapy on DPN.

3. The present language quality is not good enough and needs to be improved – numerous grammatical errors were found in the manuscript. Below are some examples of the problematic sentences.

Page 3, line 76: “In DM” – what is DM short for? Is this a typo?

Page 4, line 94:essential role in the pathogenesis of DN” – should DN be DPN?

Page 4, lines 116-118: “There are four categories of drugs are available for DPN; 1. Anticonvulsants (particularly alpha-2-delta ligand), 2. Antidepressants (mainly TCAs), serotonin-norepinephrine reuptake inhibitors (SNRIs), opiate -receptor agonists and topical agents” – when listing four categories of available drugs for DNP, please make sure the format is consistent.

Page 5, lines 130-131: “Although all these drugs used for DPN may reduce the symptoms, although no one shows effectiveness to prevent the progression of DPN.” - grammatical error.

Page 5, lines 163-164: “As a result, increases nerve conduction velocity and increase nerve vascularity [111].” - grammatical error.

Page 7, lines 214-215: “However, this study cannot be able to show the mechanism of action by which EPCs improve vascular health” - grammatical error.

Page 8, line 295-296: “after transplantation of BMSCs in STZ-induced diabetic mice, tumor formation also observed [114].” - grammatical error.

Author Response

Response to reviewer 2 comments:

The manuscript reviewed existing literatures regarding stem cell therapy for diabetic polyneuropathy (DPN). Overall, the manuscript topic is interesting in the field, but it needs improvement.

Thank you very much for raising the important concern that can make this manuscript more suitable.

Specific comments:

  1. The manuscript does not seem to be well structured. For example, table 1 on page 2 and 3 lists existing stem cell therapies on neurological diseases other than DPN, but this table is not mentioned in the text of the manuscript at all.

Answer: Thank you reviewer for noticing this error. We have cited the figures and tables in the text as suggested by reviewer.

  1. I think the manuscript will be improved if the authors could elaborate some more on the mechanisms and beneficial effects of stem cell therapy on DPN.

Answer: According to the reviewer’s suggestion, we have added some more mechanisms and beneficial effects of stem cells in DPN.

  1. The present language quality is not good enough and needs to be improved – numerous grammatical errors were found in the manuscript. Below are some examples of the problematic sentences.

Answer: Thank you very much for the suggestion. We have improved our English by editing with a native English speaker.

Page 3, line 76: “In DM” – what is DM short for? Is this a typo?

Answer: Thank you for noticing this, DM is a very commonly use the short form of diabetes mellitus both in clinical and basic science. However, we have mentioned this in the text at line 79.

Page 4, line 94: “essential role in the pathogenesis of DN” – should DN be DPN?

Answer: Thank you very much for your pointing out the typo. We have corrected it in the text.

Page 4, lines 116-118: “There are four categories of drugs are available for DPN; 1. Anticonvulsants (particularly alpha-2-delta ligand), 2. Antidepressants (mainly TCAs), serotonin-norepinephrine reuptake inhibitors (SNRIs), opiate -receptor agonists and topical agents” – when listing four categories of available drugs for DNP, please make sure the format is consistent.

Answer: Thank you very much, for your point. We have corrected the format in the text.

Page 5, lines 130-131: “Although all these drugs used for DPN may reduce the symptoms, although no one shows effectiveness to prevent the progression of DPN.” - grammatical error.

Answer: Thank you very much for your suggestions, we have correct the grammatical error.

Page 5, lines 163-164: “As a result, increases nerve conduction velocity and increase nerve vascularity [111].” - grammatical error.

Answer: Thanks for raising the comment and pointing the grammatical error. We have rephrased the sentence.

Page 7, lines 214-215: “However, this study cannot be able to show the mechanism of action by which EPCs improve vascular health” - grammatical error.

Answer: Thanks for raising the comment and pointing out the grammatical error. We have rephrased the sentence.

Page 8, line 295-296: “after transplantation of BMSCs in STZ-induced diabetic mice, tumor formation also observed [114].” - grammatical error.

Answer: Thanks for r the comment and pointing out the grammatical error. We have rephrased the sentence.

Round 2

Reviewer 2 Report

Thank the authors for providing revised manuscript. It looks much better. There are still many grammar mistakes throughout the manuscript. Please proofread it and make the corrections. Below are just some examples of grammar mistakes in the revised manuscript:

Page 3, lines 70-71: “Recent research has identified a number of pathways that contribute to the development of DPN [Figure 1]”

Page 3, lines 79-80: “all these factors are reduced leads to functional atrophy and even cellular death in the nerve”

Page 4, line 106: “Maintainin an enduring level of blood glucose level reduces the symptoms of neuropathy”

Author Response

Thank you very much for your kind review. We have gone through extensive English editing according to the reviewer’s comments. We have also resolved the grammatical errors as the reviewer suggested. The new version of the manuscript has been uploaded for your consideration.